# Salinity Tolerance of Four Hardy Ferns from the Genus *Dryopteris* Adans. Grown under Different Light Conditions

**Piotr Salachna *** and Rafał Piechocki

Department of Horticulture, West Pomeranian University of Technology, 3 Papieża Pawła VI Str., 71-459 Szczecin, Poland; rafal.piechocki@zut.edu.pl
* Correspondence: piotr.salachna@zut.edu.pl; Tel.: +48-91-4496-359

**Abstract:** Hardy ferns form a group of attractive garden perennials with an unknown response to abiotic stresses. The aim of this study was to evaluate the tolerance of three species of ferns of *Dryopteris* genus (*D. affinis*, *D. atrata* and *D. filix-mas*) and one cultivar (*D. filix-mas* cv. "Linearis-Polydactylon") to salinity and light stress. The plants were grown in full sun and shade and watered with 50 and 100 mM dm$^{-3}$ NaCl solution. All taxa treated with 100 mM NaCl responded with reduced height, leaf greenness index and fresh weight of the above-ground part. In *D. affinis* and *D. atrata* salinity caused leaf damage manifested by necrotic spots, which was not observed in the other two taxa. The effect of NaCl depended on light treatments and individual taxon. *D. affinis* and *D. atrata* were more tolerant to salinity when growing under shade. Contrary to that, *D. filix-mas* cv. "Linearis-Polydactylon" seemed to show significantly greater tolerance to this stress under full sun. Salt-treated *D. filix-mas* cv. "Linearis-Polydactylon" plants accumulated enhanced amounts of K$^+$ in the leaves, which might be associated with the taxon's tolerance to salinity. Among the investigated genotypes, *D. filix-mas* cv. "Linearis-Polydactylon" seemed the most and *D. affinis* and *D. atrata* the least tolerant to salinity and light stress.

**Keywords:** hardy ferns; salt stress; NaCl; light stress; shade; multiple stresses; abiotic stresses

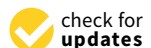



## 1. Introduction

Ornamental plants are constantly exposed to adverse effects of stress factors that disturb their growth and diminish their decorative value [1]. These factors operate both during production and after planting at the target site. The stress can be caused by an excess or deficiency of any abiotic factor [2]. Very often stresses evoked by unfavorable environmental conditions overlap with each other [3]. Most studies conducted so far have focused on the response of ornamental plants to single stress factors [1,4,5], while their response to multiple stresses experienced simultaneously is less known.

Excessive substrate salinity causing salt stress is one of the most common abiotic stresses that negatively affect the quality of ornamental plants [4]. Exceeding tolerance threshold for salt triggers numerous adverse changes in plant growth and development [6,7]. Too high salinity limits water availability, lowers turgor and leads to inhibition of cell elongation growth, tissue necrosis, yellowing, drying and falling of leaves [8]. The negative effects of salinity are due to toxicity of sodium ions that accumulate in plant tissues and disturb plant ionic balance [9]. The assessment of ion content in plants grown under salt stress is important from a cognitive and practical perspective, as the ion level might be a marker for selecting species and cultivars tolerant to elevated salinity [10,11].

Global climatic changes increase the intensity of extreme weather phenomena, including heat waves and ensuing intense solar radiation that causes light stress in plants [12,13]. Excessive solar radiation results in photoinhibition and photodestruction of pigments, and further consequences include reduced photosynthesis efficiency and extensive tissue damage [14]. Ornamental plants exposed to adverse light conditions respond with growth

disturbances and diminished decorative value [15]. There are many reports on the reaction of photophilic ornamental plants to light deficiency [16–18]. However, the effects of light stress on the quality of shade tolerant ornamental plants have not been extensively studied.

Pteridophytes (ferns) are a highly diverse plant group including at least 10,000 species growing in all climatic zones except for extremely dry or cold areas [19]. Many fern species have high decorative value and are grown in pots, gardens and as cut plants [20]. Some of them are also edible and medicinal plants [21]. In general, ferns are sensitive to salinity, excessive sunlight and water shortage. Nevertheless, a few studies reported on fern genotypes tolerant to abiotic stresses and capable of adapting to adverse habitat conditions [22–24].

Different species and cultivars collectively named hardy ferns, and comprising garden ferns with attractive foliage and habit and tolerant to frost, are gaining popularity [20]. The group includes e.g., many ornamental species of *Dryopteris* Adans genus (Dryopteridaceae family) [25]. A typical species, *Dryopteris filix-mas* L., forms loose tufts and is deemed to be one of the easiest fern species to cultivate. Of numerous morphotypes and cultivars of *D. filix-mas*, the one particularly original is cv. "Linearis-Polydactylon" forming dense tufts of upright and extremely delicate, lacy leaves. *Dryopteris atrata* has decorative, evergreen, dark-green blades sitting on petioles densely covered with dark scales. The ornamental feature of *Dryopteris affinis* are dark-green, evergreen leaves of slightly shiny, leathery texture [20,25]. Apart from their decorative qualities, *D. affinis*, *D. atrata* and *D. filix-mas* are also sources of valuable metabolites for phytomedicine [26]. Wider dissemination of *Dryopteris* species and cultivars is prevented by a lack of detailed data on the methods of their cultivation. Moreover, there is little information on *Dryopteris* resistance to environmental stresses.

The stress response of ornamental seed plants is relatively well researched [1,14] but the effects of stress factors on growth and ornamental value of spore plants are much less known. To fill in the gaps in our knowledge, this study investigated the response of selected hardy ferns to salt stress, light stress and a combination of both. We assessed the effects of two salinity levels (50 and 100 mM NaCl) and two light treatments (sun and shade) on morphological traits, visual score, leaf greenness index and leaf content of $Na^+$, $K^+$ and $Ca^{2+}$ in four taxa of *Dryopteris* genus. We hypothesized that the tested hardy ferns may include taxa tolerant to environmental stresses.

## 2. Materials and Methods

### 2.1. Plant Material

The study involved three species of hardy ferns belonging to *Dryopteris* genus (*D. affinis*, *D. atrata* and *D. filix-mas*) and one cultivar of this genus (*D. filix-mas* "Linearis-Polydactylon"). The plants were propagated in vitro and acclimatized in pots (0.5 $dm^3$ capacity) in a greenhouse at a horticultural farm (Rzgów, Poland). Each plant had 5–7 leaves and a very well developed rhizome clump.

### 2.2. Culture Conditions

The plants were planted on 15 July 2019. Each round, a black PVC pot of 1.7 $dm^3$ capacity harbored a single plant and was filled with peat substrate (pH 6.0) supplemented with PG Mix fertilizer (Yara, Poland) at a dose of 1.0 kg $m^{-3}$. The fertilizer contained: 5.5% N-$NO_3$, 8.5% N-$NH_4$, 16% $P_2O_5$, 18% $K_2O$, 0.8% MgO, 19% $SO_3$, 0.03% B, 0.12% Cu, 0.09% Fe, 0.16% Mn, 0.20% Mo, and 0.04% Zn. The pots were placed on white nursery mats in an unheated plastic tunnel of the area of 225 $m^2$, and covered with a double layer of inflated poly film (lat. 53°25′ N, long. 14°32′ E; elevation, 25 m). The plants were watered every three days with tap water of pH 6.4, electrolytic conductivity 0.64 mS $cm^{-1}$ containing (mg $dm^{-3}$) 6.2 $K^+$, 98 $Ca^{2+}$ and 25 $Na^+$. The tunnel's roof ventilation opened when the temperature inside exceeded 18 °C. Air temperature and relative humidity were recorded with a portable USB data logger. The average monthly maximum/minimum air temperature and average maximum/minimum relative humidity (RH) in the plastic house were, respectively: July 28.6 °C/15.1 °C, RH 93.3%/44.9%;

August 29.5 °C/15.2 °C, RH 94.5%/44.8%; September 22.8 °C/11.3 °C, RH 96.5%/56.1%; and October 21.9 °C/9.2 °C, RH 91.6%/67.0%.

### 2.3. Experimental Design and Treatments

Form 5 August 2019 until the end of the trial (23 October 2019), half plants of each fern taxon were grown in a tunnel in full sun, and the other half of the plants were grown in a tunnel under shading screens (a highly reflective aluminized shade fabric). Measurements with a Radiometer-Fotometr RF-100 (Sonopan, Białystok, Poland) determined the photosynthetic photon flux density (PPFD) on a cloudless day of 4 August 2019 at 609.1 $\mu$mol m$^{-2}$ s$^{-1}$ for full sun and 151 $\mu$mol m$^{-2}$ s$^{-1}$, i.e., 24% of this value, in shade.

Starting on 5 August 2019, plants at the two sites (full sun and shade) were watered four times, i.e., every five days with a solution of sodium chloride (NaCl) pure p.a. 99.9% (Chempur, Poland). NaCl concentration was either 50 or 100 mM, and each plant was provided with 100 mL of the solution per watering. The control plants were irrigated with tap water. NaCl concentrations were selected based on a study by Bogdanovic et al. [24]. After the last dose of NaCl was applied, all plants were watered with tap water until the end of the experiment.

The experimental design was a sub-block one, with three repetitions per combination and nine plants per repetition.

### 2.4. Assessment of Greenness Index, Ornamental Value and Morphological Features

On the last day of the experiment we measured leaf greenness index in Soil Plant Analysis Development (SPAD) units with a Chlorophyll Meter SPAD-502 (Minolta, Japan). The measurements were conducted between 10.00 a.m. and noon and included three fully developed leaves without any signs of necrosis, located in the middle of the plant. Three readings were taken per each leaf. Nine plants were assessed per each combination.

To determine the decorative value of the plants, five researchers conducted a bonitation assessment (visual score) by rating all plants according to a five-point scale, where 1 meant low attractiveness, expressed as insufficient foliage, poor growth and unattractive habit, and 5 meant the maximum decorative value manifested in vigorous growth, attractive habit and healthy foliage.

All plants in all combinations were also assessed for their height (from the soil surface to the tip of the tallest leaf) and fresh weight of the above-ground part cut at the substrate level in the pots.

### 2.5. Analysis of Na$^+$, K$^+$ and Ca$^{2+}$ Content

To determine the content of Na$^+$, K$^+$ and Ca$^{2+}$, the collected leaves were rinsed twice with deionized water, blotted dry, placed into brown paper bags and left in an oven at 65 °C for 72 h. Dried material was pulverized into particles of diameter below 1 mm, and wet mineralized in 17 mL of 96–97% H$_2$SO$_4$ per 2.0 g of the material. The ion content was determined by the flame photometry on a flame photometer AFP-100 (Biotech Engineering Management, Nicosia, Cyprus, as described by Ostrowska [27]. Each mineral was determined in three analytical replicates per treatment.

### 2.6. Statistical Analysis

The experimental data were statistically analyzed by means of a variance analysis for two-factor (salinity and light) experiments in Statistica Professional 13.3 package (TIBCO Software, Palo Alto, CA, USA). Date were analyzed separately for each taxon. The multiple comparison procedure based on the Tukey's HSD post-hoc test with the significance level $p \leq 0.05$ was used to identify differences between the means.

## 3. Results

### 3.1. Overall Effects of Salinity Treatments

Salinity stress strongly affected plant height, leaf greenness index (SPAD), fresh weight of the above-ground parts (Figure 1a–c) and visual score (Table 1) of all investigated fern taxa. Plants of all combinations survived the salt stress. NaCl at 50 and 100 mM caused a clear plant height reduction in *D. atrata*, *D. affinis*, and *D. filix-mas* whereas *D. filix-mas* cv. "Linearis-Polydactylon" responded this way only to 100 mM NaCl. SPAD index in *D. atrata*, *D. affinis* and *D. filix mas* dropped with growing salt concentration, and it was also reduced in *D. filix-mas* cv. "Linearis-Polydactylon" but did not depend on NaCl levels. In *A. atrata* and *A. affinis* the drop in fresh weight of the above-ground parts was more intense at higher NaCl concentration. Fresh weight of *D. filix-mas* plants was also lower under salt stress but no significant differences were spotted for 50 and 100 mM NaCl. In *D. filix-mas* cv. "Linearis-Polydactylon" plants the decrease in fresh weight was only visible at 100 mM NaCl. Salinity considerably affected the visual score of *A. atrata* and *D. affinis* in a concentration dependent way. In *D. filix* species and its cultivar "Linearis-Polydactylon" the visual score was also lower in the presence of salt but there was no difference between NaCl concentrations.

(**a**)

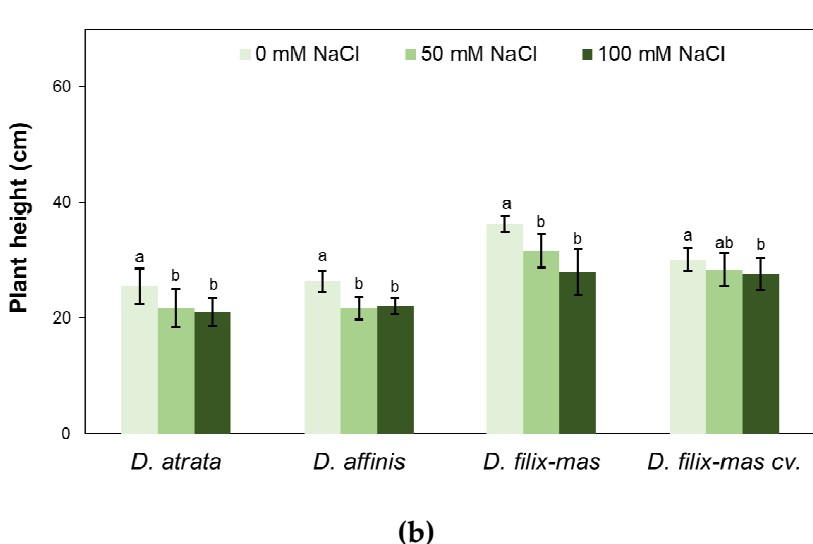

(**b**)

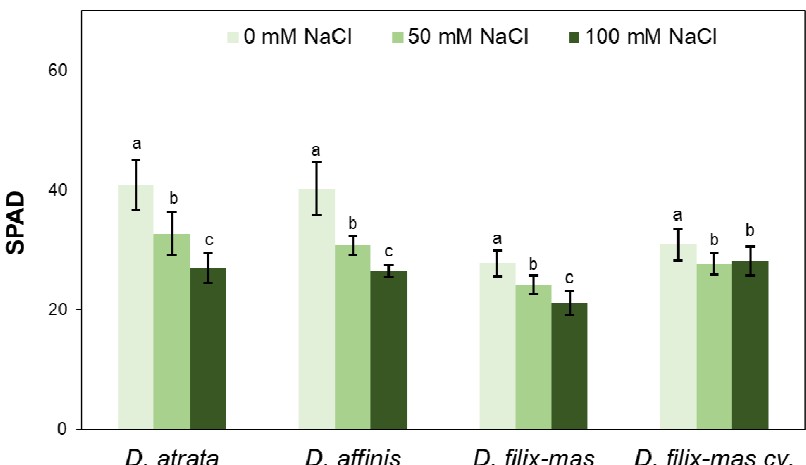

**Figure 1.** *Cont.*

**(c)**

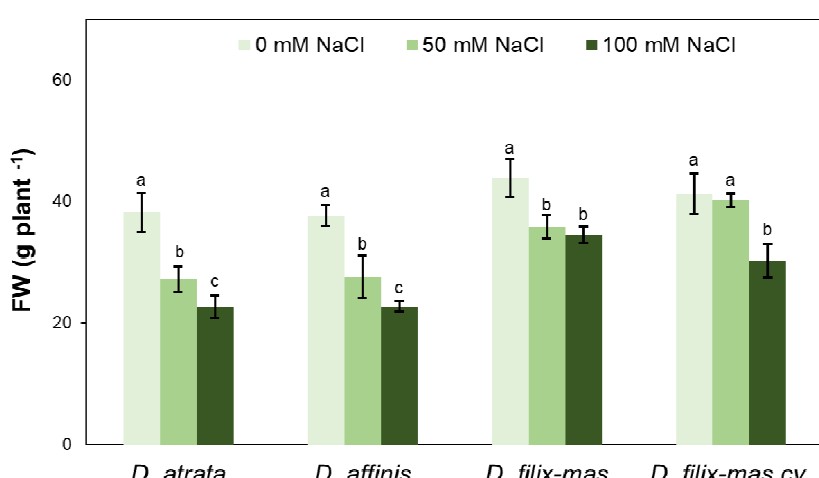

**Figure 1.** Effect of salinity on plant height (**a**), leaf greenness index (Soil Plant Analysis Development (SPAD)) (**b**) and fresh weight of the above-ground part (**c**) of *D. affinis*, *D. atrata*, *D. filix-mas* and *D. filix-mas* cv. Linearis-Polydactylon (*D. filix-mas* cv.). Data are mean ± SD. Different letters indicate significant differences for $p \leq 0.05$.

**Table 1.** Effect of salinity on visual score of *D. affinis*, *D. atrata*, *D. filix-mas* and *D. filix-mas* cv. Linearis-Polydactylon (*D. filix-mas* cv.). Data are expressed as mean and standard deviation (±SD).

| Salinity | Species/Cultivar | | | |
|---|---|---|---|---|
| (mM NaCl) | *D. atrata* | *D. affinis* | *D. filix-mas* | *D. filix-mas* cv. |
| 0 | 4.5 ± 0.6a [1] | 4.9 ± 0.1a | 4.5 ± 0.6a | 4.7 ± 0.4a |
| 50 | 2.6 ± 0.6b | 3.8 ± 1.2b | 3.7 ± 0.5b | 4.3 ± 0.3b |
| 100 | 2.0 ± 1.1c | 3.2 ± 0.6c | 3.5 ± 0.5b | 4.4 ± 0.5b |

[1] Means not marked with the same letter are significantly different at $p \leq 0.05$.

In all fern taxa, salinity significantly increased the leaf content of $Na^+$ with increasing rates of NaCl (Table 2). Salt treatment resulted in a drop of $K^+$ levels in *D. atrata* and its surge in *D. filix-mas* cv. "Linearis-Polydactylon" at both NaCl levels. Plants of both taxa exposed to the higher NaCl dose (100 mM) accumulated lower content of $Ca^{2+}$.

**Table 2.** Effect of salinity on $Na^+$, $K^+$ and $Ca^{2+}$ content (expressed in % dry weight) in leaves of *D. affinis*, *D. atrata*, *D. filix-mas* and *D. filix-mas* cv. Linearis-Polydactylon (*D. filix-mas* cv.). Data are expressed as mean and standard deviation (±SD).

| Ion Content (% DW) | Salinity (mM NaCl) | Species/Cultivar | | | |
|---|---|---|---|---|---|
| | | *D. atrata* | *D. affinis* | *D. filix-mas* | *D. filix-mas* cv. |
| | 0 | 0.24 ± 0.03c [1] | 0.19 ± 0.07c | 0.24 ± 0.06c | 0.27 ± 0.06c |
| $Na^+$ | 50 | 0.41 ± 0.06b | 0.73 ± 0.09b | 0.67 ± 0.10b | 0.46 ± 0.10b |
| | 100 | 1.53 ± 0.13a | 0.86 ± 0.06a | 0.89 ± 0.15a | 1.12 ± 0.06a |
| | 0 | 1.62 ± 0.03a | 1.25 ± 0.16 | 1.28 ± 0.12 | 1.08 ± 0.09b |
| $K^+$ | 50 | 1.22 ± 0.15b | 1.23 ± 0.09 | 1.35 ± 0.11 | 1.46 ± 0.24a |
| | 100 | 1.26 ± 0.07b | 1.41 ± 0.16 | 1.38 ± 0.06 | 1.38 ± 0.17a |
| | 0 | 0.97 ± 0.06a | 0.88 ± 0.16 | 0.72 ± 0.04 | 0.74 ± 0.14a |
| $Ca^{2+}$ | 50 | 0.93 ± 0.10a | 0.91 ± 0.07 | 0.80 ± 0.07 | 0.69 ± 0.14ab |
| | 100 | 0.76 ± 0.04b | 0.93 ± 0.08 | 0.75 ± 0.11 | 0.56 ± 0.11b |

[1] Means not marked with the same letter are significantly different at $p \leq 0.05$.

### 3.2. Overall Effects of Light Treatments

The effects of light conditions on plant height, leaf greenness index (SPAD), fresh weight of the above-ground parts (Figure 2a–c) and visual score (Table 3) was variable and taxon-dependent. *D. atrata* plants growing in full sun were lower, had smaller fresh weight of the above-ground parts and a lower SPAD index and visual score than shaded plants. Similarly, *D. affinis* plants grown under full sun had lower fresh weight and reduced SPAD index and visual score. In *D. filix-mas*, light conditions did not affect fresh weight or SPAD index but resulted in differences in plant height and visual score. *D. filix-mas* plants growing in full sun were higher but those growing in the shade had higher visual score. *D. filix-mas* cv. "Linearis-Polydactylon" plants in high light reached greater SPAD index and higher height than their shaded counterparts. We detected no effects of light availability on the content of $Na^+$, $K^+$ or $Ca^{2+}$ in all tested ferns ($p > 0.05$, results not shown).

**(a)**

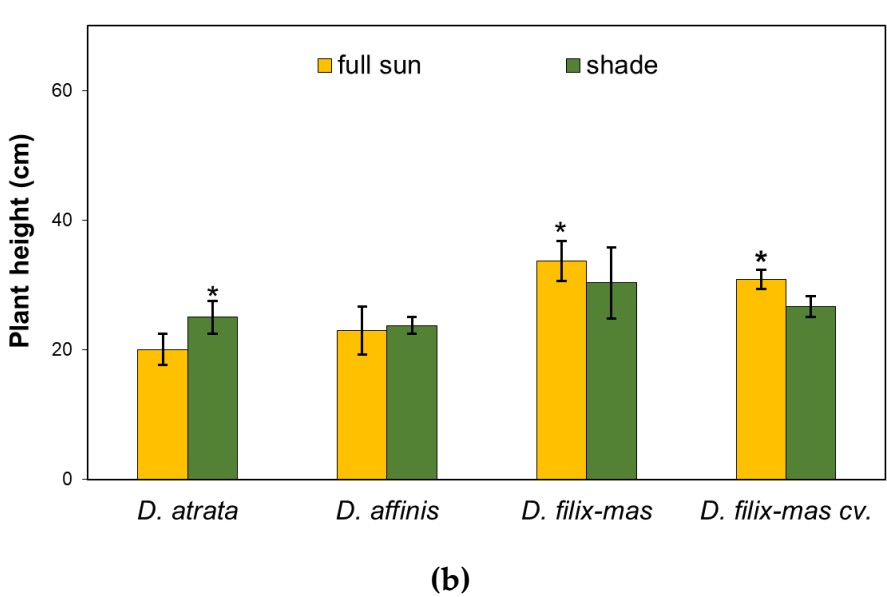

**(b)**

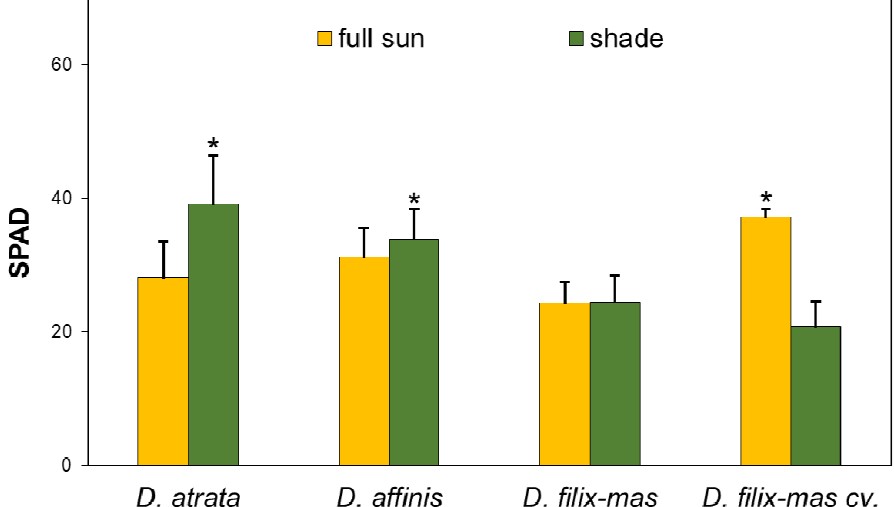

**Figure 2.** *Cont.*

**(c)**

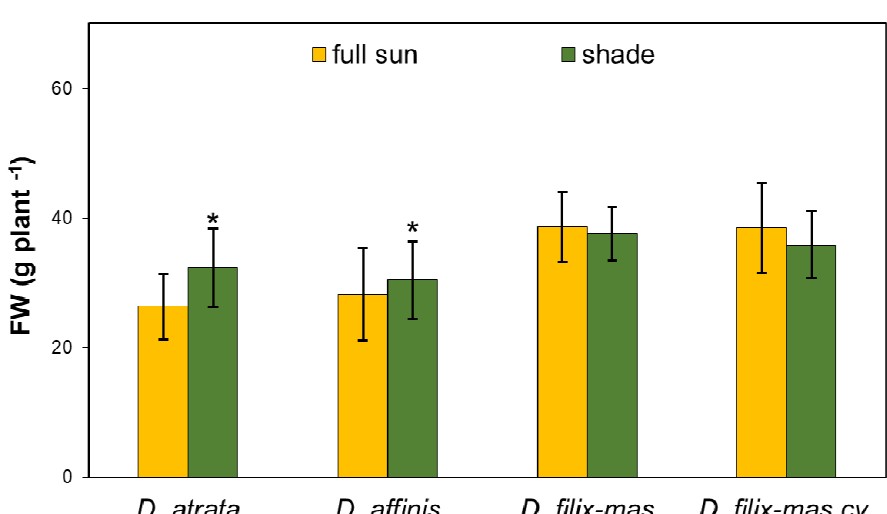

**Figure 2.** Effect of light conditions on plant height (**a**), leaf greenness index (SPAD) (**b**) and fresh weight of the above-ground part (**c**) of *D. affinis*, *D. atrata*, *D. filix-mas* and *D. filix-mas* cv. Linearis-Polydactylon (*D. filix-mas* cv.). Data are mean ± SD. Asterisks mark indicate significant differences for $p \leq 0.05$.

**Table 3.** Effect of light conditions on visual score of *D. affinis*, *D. atrata*, *D. filix-mas* and *D. filix-mas* cv. Linearis-Polydactylon (*D. filix-mas* cv.). Data are mean ± SD.

| Light Conditions | Species/Cultivar | | | |
| --- | --- | --- | --- | --- |
| | *D. atrata* | *D. affinis* | *D. filix-mas* | *D. filix-mas* cv. |
| Full sun | 2.3 ± 1.3b [1] | 3.4 ± 1.0b | 3.4 ± 0.5b | 4.5 ± 0.5 |
| Shade | 3.7 ± 1.0a | 4.6 ± 0.6a | 4.3 ± 0.5a | 4.4 ± 0.4 |

[1] Means not marked with the same letter are significantly different at $p \leq 0.05$.

### 3.3. Combined Effects of Salinity and Light Treatment

The effects of salt stress on plant height, leaf greenness index (SPAD), fresh weight of the above-ground parts (Figure 3a–c) and visual score (Table 4) depended on light condition and taxon. In *D. atrata* and *D. affinis* salinity reduced plant height considerably stronger in plants growing in full-sun than in shade. In *D. filix-mas* and cv. "Linearis-Polydactylon" NaCl only slightly diminished plant height under both light conditions. Exposure to both concentrations of salt resulted in a decrease of fresh weight of *D. atrata*, *D. affinis* and *D. filix-mas* in both light treatments, whereas *D. filix-mas* cv. "Linearis-Polydactylon" responded with a drop in fresh weight, both in the sun and in the shade, only to 100 mM NaCl. SPAD greenness index decreased in *D. atrata* and *D. affinis* with increasing concentration of NaCl both under full sun and shade treatments. In *D. filix-mas*, its salinity-triggered reduction was only perceived in low light intensity. In salt-exposed *D. filix-mas* cv. "Linearis-Polydactylon" plants SPAD value declined in the shade but grew in the sun. Control (no salt) and shaded plants of *D. atrata*, *D. affinis* and *D. filix* achieved the highest visual score. In *D. filix-mas* cv. "Linearis-Polydactylon" the most decorative plants were those growing without NaCl pressure in the full sun. Interestingly, we found no leaf discoloration or necrosis in *D. filix* and *D. filix-mas* cv. "Linearis-Polydactylon" exposed to salt under both light conditions. Salt-exposed plants of *D. atrata* and *D. affinis* responded with leaf margin chlorosis and necrosis, particularly at 100 NaCl mM and under full sun (Figure 4).

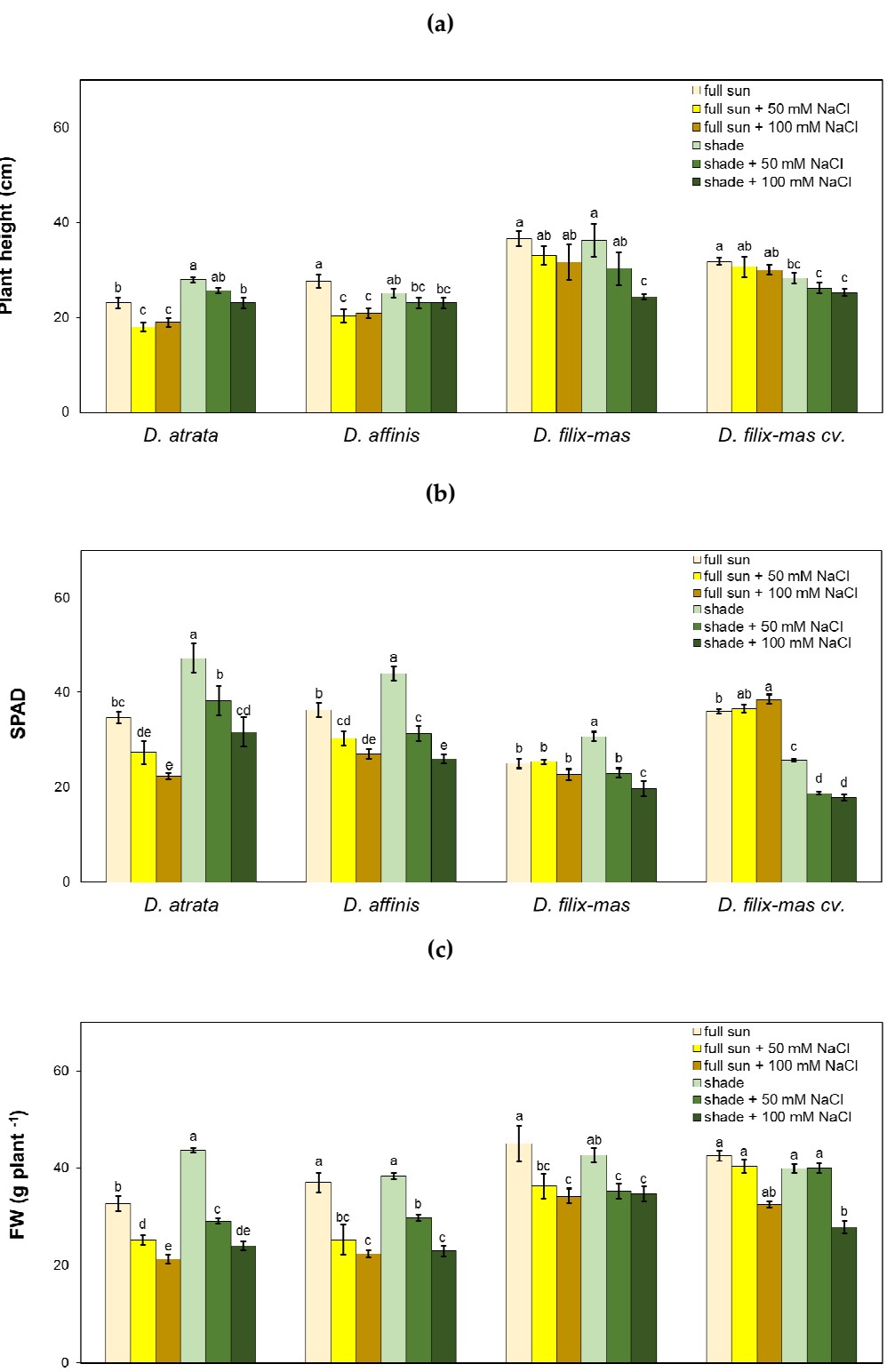

**Figure 3.** Effect of light conditions and salinity on plant height (**a**), leaf greenness index (SPAD) (**b**) and fresh weight of the above-ground part (**c**) of *D. affinis*, *D. atrata*, *D. filix-mas* and *D. filix-mas* cv. Linearis-Polydactylon (*D. filix-mas* cv.). Data are mean ± SD. Different letters indicate significant differences for $p \leq 0.05$.

**Table 4.** Effect of light conditions and salinity on visual score of *D. affinis*, *D. atrata*, *D. filix-mas* and *D. filix-mas* cv. Linearis-Polydactylon (*D. filix-mas* cv.). Data are mean ± SD.

| Light Conditions | Salinity (mM NaCl) | Species/Cultivar | | | |
|---|---|---|---|---|---|
| | | *D. atrata* | *D. affinis* | *D. filix-mas* | *D. filix-mas* cv. |
| Full sun | 0 | 3.9 ± 0.1b [1] | 4.8 ± 0.1a | 3.9 ± 0.1b | 5.00 ± 0.00a |
| | 50 | 2.0 ± 0.0d | 2.8 ± 0.1c | 3.3 ± 0.6bc | 4.55 ± 0.19bc |
| | 100 | 1.0 ± 0.0e | 2.6 ± 0.2c | 3.1 ± 0.1c | 3.96 ± 0.07d |
| Shade | 0 | 5.0 ± 0.0a | 5.0 ± 0.0a | 5.0 ± 0.0a | 4.29 ± 0.25cd |
| | 50 | 3.1 ± 0.2c | 4.9 ± 0.1a | 4.0 ± 0.0b | 4.07 ± 0.13d |
| | 100 | 3.1 ± 0.1c | 3.8 ± 0.1b | 3.9 ± 0.1b | 4.85 ± 0.13ab |

[1] Means not marked with the same letter are significantly different at $p \leq 0.05$.

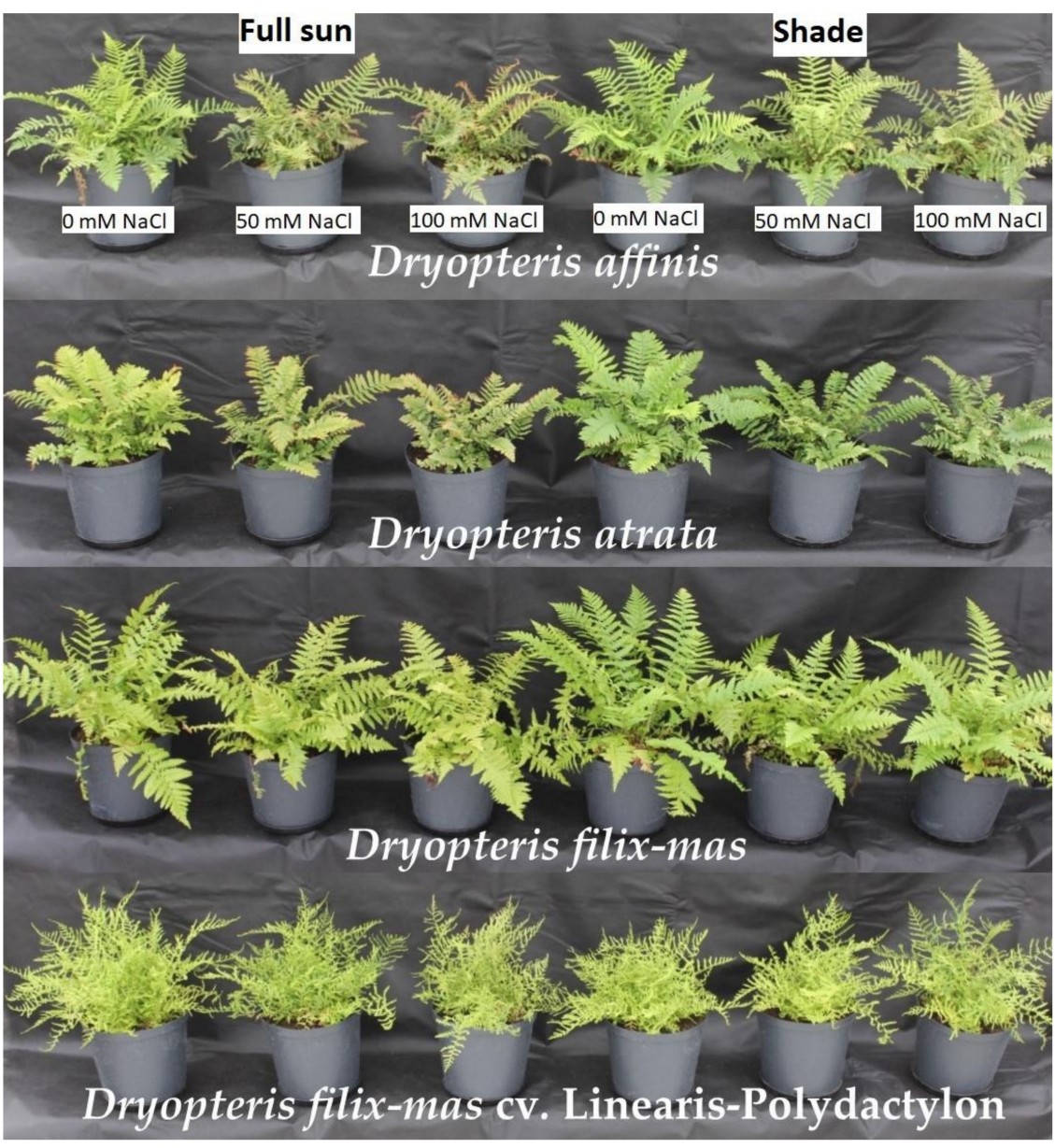

**Figure 4.** Effect of light conditions and salinity on growth of *Dryopteris affinis*, *Dryopteris atrata*, *Dryopteris filix-mas* and *Dryopteris filix-mas* cv. Linearis-Polydactylon. Left to right: full sun; full sun + 50 mM NaCl; full sun + 100 mM NaCl; shade; shade + 50 mM NaCl and shade + 100 mM NaCl.

The greatest content of Na$^+$ was found in all ferns treated with 100 mM NaCl, irrespective of light conditions. In *D. atrata* plants cultivated under full sun and shade, salinity at 50 and 100 mM NaCl resulted in lowering K$^+$ content. In *D. filix-mas* cv. "Linearis-Polydactylon" NaCl at 50 and 100 mM boosted K$^+$ levels irrespective of light intensity. *D. atrata* treated with 100 mM NaCl and *D. filix-mas* cv. "Linearis-Polydactylon" treated with 50 mM NaCl exposed to the shade accumulated smaller amounts of Ca$^{2+}$ (Table 5).

**Table 5.** Effect of light conditions and salinity (50 and 100 mM NaCl) on Na$^+$, K$^+$ and Ca$^{2+}$ content (expressed in % dry weight) in leaves of *D. affinis*, *D. atrata*, *D. filix-mas* and *D. filix-mas* cv. Linearis-Polydactylon (*D. filix-mas* cv.). Data are mean $\pm$ SD.

| Light Conditions | Salinity (mM NaCl) | Species/Cultivar | | | |
|---|---|---|---|---|---|
| | | *D. atrata* | *D. affinis* | *D. filix-mas* | *D. filix-mas* cv. |
| | | **Na$^+$** | | | |
| | 0 | 0.22 ± 0.02b [1] | 0.21 ± 0.09b | 0.24 ± 0.04b | 0.27 ± 0.07bc |
| Full sun | 50 | 0.38 ± 0.07b | 0.69 ± 0.09a | 0.62 ± 0.10a | 0.43 ± 0.13bc |
| | 100 | 1.60 ± 0.15a | 0.84 ± 0.04a | 0.88 ± 0.09a | 1.08 ± 0.08a |
| | 0 | 0.27 ± 0.02b | 0.17 ± 0.06b | 0.23 ± 0.09b | 0.26 ± 0.06c |
| Shade | 50 | 0.43 ± 0.06b | 0.77 ± 0.08a | 0.72 ± 0.08a | 0.49 ± 0.08b |
| | 100 | 1.45 ± 0.07a | 0.88 ± 0.08a | 0.91 ± 0.22a | 1.15 ± 0.02a |
| | | **K$^+$** | | | |
| | 0 | 1.64 ± 0.04a | 1.30 ± 0.10 | 1.33 0.15 | 1.09 ± 0.15b |
| Full sun | 50 | 1.12 ± 0.13c | 1.26 ± 0.12 | 1.33 0.15 | 1.49 ± 0.29a |
| | 100 | 1.32 ± 0.02b | 1.38 ± 0.08 | 1.37 0.08 | 1.29 ± 0.14ab |
| | 0 | 1.61 ± 0.02a | 1.19 ± 0.21 | 1.23 0.06 | 1.08 ± 0.03b |
| Shade | 50 | 1.33 ± 0.06b | 1.20 ± 0.04 | 1.37 0.08 | 1.43 ± 0.24a |
| | 100 | 1.20 ± 0.03bc | 1.44 ± 0.24 | 1.39 0.05 | 1.46 ± 0.17a |
| | | **Ca$^{2+}$** | | | |
| | 0 | 0.94 ± 0.05ab | 0.99 ± 0.16 | 0.72 ± 0.03 | 0.65 ± 0.08ab |
| Full sun | 50 | 0.88 ± 0.08a–c | 0.93 ± 0.04 | 0.75 ± 0.05 | 0.65 ± 0.06ab |
| | 100 | 0.78 ± 0.03bc | 0.92 ± 0.11 | 0.76 ± 0.08 | 0.65 ± 0.14ab |
| | 0 | 0.99 ± 0.06a | 0.76 ± 0.07 | 0.73 ± 0.06 | 0.83 ± 0.14a |
| Shade | 50 | 0.98 ± 0.10a | 0.89 ± 0.09 | 0.86 ± 0.05 | 0.47 ± 0.06b |
| | 100 | 0.75 ± 0.05c | 0.95 ± 0.05 | 0.73 ± 0.15 | 0.74 ± 0.15ab |

[1] Means not marked with the same letter are significantly different at $p \leq 0.05$.

## 4. Discussion

During their growth and development plants are exposed to different environmental stresses, the effects of which are often synergistic, and their combined outcome is considerably more powerful than that of individual stress factors [28,29]. Understanding the response of individual genotypes to adverse environmental conditions allows for proper selection of tolerant and resistant plants [7,10,30]. Most studies on the effects of stressful conditions have been carried out on flower ornamentals, while the group of leaf ornamental plants has so far received very little attention. The aim of this work was to investigate the response of four ferns of *Dryopteris* genus, generally considered as shade plants, to multi-stress in the form of salinity and high light intensity.

Most plants exposed to excessive salinity limit the elongation growth of cells, which results in reduced growth and biomass production [31,32]. Salt stress often diminishes visual quality of plants by evoking brownish necrosis of leaves [4,8]. In our study, salinity also inhibited growth, reduced fresh weight of the above-ground part and lowered the bonitation score of the investigated ferns, and intensity of these effects depended on the taxon and light conditions (Figure 3, Table 4). The species of *D. affinis* and *D. atrata* turned out the most sensitive to salt and they demonstrated leaf margin browning and drying (Figure 4). Negative effects of salinity on the growth and quality of *D. affinis* and *D. atrata* were particularly visible under full sun. In the shade, the stress affected growth and

ornamental value of *D. affinis* and *D. atrata* to a lesser degree. Our results confirmed shade affinity of *D. affinis* and *D. atrata,* and what is more, shade mitigated negative effects of salt in these species. Similarly, Medina et al. [33] showed that a halophytic fern *Acrostichum aureum* was much more tolerant to salt stress when growing in the shade than in the sun. In *Hibiscus tiliaceus* Hau, cultivated under different light conditions, salinity caused stronger total biomass reduction in plants growing in 90% shade than in full sun and 50% shade [34]. In a heliophilous species *Vicia faba*, the toxic effects of salinity were more considerably alleviated by higher than lower light intensity [35]. In our study, the same relationship was demonstrated in *D. filix-mas* cv. "Linearis-Polydactylon", as salinity experienced by plants growing under full sun did not reduce fresh weight of their above-ground parts. *D. filix-mas* cv. "Linearis-Polydactylon" plants cultivated in shade were the smallest and had the lowest fresh weight. We noticed no clear effects of light conditions on fresh weight of *D. filix-mas* but plants growing under full sun demonstrated lower ornamental value than those under low light intensities. As shown by Ure [36], *D. filix-mas* tolerates a wide range of light/shade levels.

In sensitive species salt stress reduces chlorophyll content, while in tolerant ones the pigment level remains unchanged or may even rise [37,38]. Our study assessed leaf greenness index that correlates with chlorophyll content [39]. We found a negative effect of salt stress on leaf greenness in all shaded ferns (Figure 3b). In full sun SPAD index was clearly lowered in all ferns exposed to salinity, except for *D. filix-mas* cv. "Linearis-Polydactylon", where NaCl slightly enhanced SPAD value. Bogdanovic et al. [24] tested the response of *Asplenium viride* Britton, *Ceterach officinarum* DC and *Phyllitis scolopendrium* (L.) Newmann to salt stress (0–500 mM NaCl) in vitro and found that high concentrations of NaCl (250 mM and above) drastically lowered total chlorophyll content in all species, while low concentrations (50 and 100 mM NaCl) enhanced the pigment content in *A. viride* and *C. officinarum*. Experimentally demonstrated stimulating effect of salinity on the greenness index of *D. filix-mas* cv. "Linearis-Polydactylon" may indicate that this cultivar grown under full sun is tolerant to increased salinity.

NaCl evoked salinity may disturb ion homeostasis and, consequently, disrupt the physiological processes [40]. Usually, excessive content of $Na^+$ results in deficiency of $K^+$ and $Ca^+$ [37,41]. There are, however, also contradictory data suggesting that salinity causes increased accumulation of $K^+$ [31] and $Ca^{2+}$ [42]. Potassium and calcium ions regulate activity of numerous enzymes [43,44], and their deficiency decreases plant stress tolerance [45]. In our experiment, the content of $Na^+$ rose in all taxa exposed to salinity (Table 2) due to using NaCl solution as a stress factor. Enhanced content of $Na^+$, as a major solute responsible for increased osmotic pressure of the cell sap, was also observed in salt-treated fern *A. aureum* [33]. A particularly interesting outcome of this study was a boost in $K^+$ content in *D. filix-mas* cv. "Linearis-Polydactylon". Similarly, Vogelien et al. [46] showed that a mutant of *Ceratopteris richardii stl2*, relatively tolerant to NaCl, accumulated greater amounts of $K^+$ when grown on NaCl-supplemented medium than other fern genotypes. As mentioned earlier, despite NaCl treatment *D. filix-mas* cv. "Linearis-Polydactylon" maintained its high bonitation score and greenness index, which may indicate its tolerance to the applied NaCl doses. Furthermore, an increased content of $K^+$ may suggest a role of these ions in plant adaptation to salt stress. A precise marker of salt stress in *A. aureum* was the content of cyclitol d-1-*O*-methyl-muco-inositol, a cytoplasmic compatible solute [33], while other ferns, i.e., *A. viride*, *C. officinarum* and *P. scolopendrium* responded to NaCl with a shift in total leaf phenolic content [24]. The mechanisms of plant tolerance to stress are highly complex and multidirectional. Therefore, to better understand fern tolerance to salinity, we need further studies, particularly on the level of oxidative stress and compatible solutes that protect protein structure and biological membranes against negative effects of excessive salt concentrations.

## 5. Conclusions

From among four investigated fern taxa, *D. filix-mas* cv. "Linearis-Polydactylon" showed the greatest tolerance to salt stress. Despite salinity, plants of this cultivar maintained intense, green coloration of leaves assessed by SPAD greenness index, high visual score and demonstrated increased accumulation of $K^+$ in the leaves. *D. affinis* and *D. atrata* turned out sensitive to salinity, as manifested in leaf necrosis. The effects of salt stress on plant growth depended on light condition and taxon; *D. filix-mas* cv. "Linearis-Polydactylon" plants were more tolerant to salinity when growing under full sun, and *D. affinis* and *D. atrata* showed better tolerance to NaCl under shade. Our knowledge on the impact of abiotic stresses on the growth of ornamental garden plants from the fern group is scarce, which is why these findings seem important and may serve as practical recommendations for the selection of fern species intended for areas exposed to environmental stresses.

**Author Contributions:** Conceptualization, methodology, formal analysis, writing and visualization, P.S.; investigation and data curation, R.P.; All authors have read and agreed to the published version of the manuscript.

**Funding:** This research received no external funding.

**Institutional Review Board Statement:** Not applicable.

**Informed Consent Statement:** Not applicable.

**Data Availability Statement:** Data sharing not applicable.

**Conflicts of Interest:** The authors declare no conflict of interest.

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
