# Peer review of "Salinity Tolerance of Four Hardy Ferns from the Genus Dryopteris Adans. Grown under Different Light Conditions"

_agronomy, doi:10.3390/agronomy11010049_

Round 1
Reviewer 1 Report
The authors have assessed the effect of salinity and light intensity on hardy fern growth, leaf greenness, ornamental value, and some minerals. Overall, the study is interesting and provides relevant insights about the hardy fern resistance to unfavorable environmental factors. Still, there are a number of shortcomings that need to be addressed.
Abstract: The number of words should be 200 maximum, please revise to reduce the length of your abstract.
Materials and methods:
P2 line 81: what does it mean P9?
P2 line 82: few leaves? how many? indicate a specific number of leaves.
The section 2.3 Experimental design and treatments should be overwritten. It would be much clearer if the experimental scheme were presented graphically. It is now unclear whether the experiments were performed simultaneously or whether salinity and lighting were initiated at the same time and then the plants were exposed to the complex effect.
What was the size of the plants when the effects were initiated? Was PPFD measured continuously every day or just one day (4 August 2019)? What was used to design the shading screens? Did you have the same light spectra under the full sun and under the screens? Indicate clearly how long the effect lasted. How many times the experiment was repeated in time?
P3 line 113-114: whether measurements were made on fully developed leaves?
P3 line 12-121:Did you count the leaves of the plants? Have you noticed the growth of new leaves, were there differences between the variants?
P3 line 128: authors indicate three repetitions, whether it was biological or analytical replicates?
Results
In the text, the authors use "growth" when describing the height of the plant. Plant height must be used.
P4 line 140: the same effects were also determined on D. filix-mas, see results in Fig 1.
P4 line 150-154: Review the description with the data in Table 2.
In all Figures the cultivar should be indicated on the x-axis.
Table 2: carefully check the data and letters indicating significant differences.
In the text, instead of bonitation score use the visual score.
P6 line 174-175: the data could be presented if even significant differences were not detected.
Table 3 Effects of light or salinity?
Page 7 line 201-206: Carefully check the description and the data in Table 5.
Figure 4. For clarity, the "effects" could be indicated on the picture.
Discussion
P10 line 229-236 deals more likely to a review of literature. Please reduce this (or move most in your introduction). The aim of the work should be stated only in the introduction.
P 11 line 283 specify exactly to which stress.
Author Response
Thank you very much for your time spent on a careful and detailed revision of our manuscript. We are grateful for numerous comments and remarks that made us reconsider many fundamental issues. Invaluable content and style related corrections allowed us to avoid multiple mistakes. In general, the review let us considerably improve our manuscript. Below you will find our answers to all the remarks. We hope our explanations are comprehensive and will dispel any possible doubts.
The authors have assessed the effect of salinity and light intensity on hardy fern growth, leaf greenness, ornamental value, and some minerals. Overall, the study is interesting and provides relevant insights about the hardy fern resistance to unfavorable environmental factors. Still, there are a number of shortcomings that need to be addressed.
Abstract: The number of words should be 200 maximum, please revise to reduce the length of your abstract.
Answer: The length of abstract reduced.
Materials and methods:
P2 line 81: what does it mean P9?
Answer: P9 is an abbreviation for pot size. Abbreviation has been removed.
P2 line 82: few leaves? how many? indicate a specific number of leaves.
Answer: The number of leaves is given.
The section 2.3 Experimental design and treatments should be overwritten. It would be much clearer if the experimental scheme were presented graphically. It is now unclear whether the experiments were performed simultaneously or whether salinity and lighting were initiated at the same time and then the plants were exposed to the complex effect.
Answer: This part of the work has been revised to dispel doubts
What was the size of the plants when the effects were initiated? Was PPFD measured continuously every day or just one day (4 August 2019)? What was used to design the shading screens? Did you have the same light spectra under the full sun and under the screens? Indicate clearly how long the effect lasted. How many times the experiment was repeated in time?
Answer: Only homogeneous plants were selected for the research. PPFD measurements were not performed daily. The type of shading was determined. Not studied light spectra under the full sun and under the screens. The paper indicates exactly how long the research was conducted and how many repetitions.
P3 line 113-114: whether measurements were made on fully developed leaves?
Answer: Only fully expanded leaves were selected for measurements
P3 line 12-121:Did you count the leaves of the plants? Have you noticed the growth of new leaves, were there differences between the variants?
Answer: Plants had leaves of various sizes in various stages of development and therefore I did not provide the number of leaves in this study.
P3 line 128: authors indicate three repetitions, whether it was biological or analytical replicates?
Answer: Three readings of the mineral content were obtained. This has been specified.
Results
In the text, the authors use "growth" when describing the height of the plant. Plant height must be used.
Answer: Corrected in line with comments.
P4 line 140: the same effects were also determined on D. filix-mas, see results in Fig 1.
Answer: Thank you for checking very carefully. Corrected in line with comments
P4 line 150-154: Review the description with the data in Table 2.
Answer: Thank you for checking very carefully. Corrected in line with comments.
In all Figures the cultivar should be indicated on the x-axis.
Answer: This cultivar has a very long name and to make the charts transparent I limited its caption. The full cultivar name is given in the diagram description.
Table 2: carefully check the data and letters indicating significant differences.
Answer: Corrected in line with comments.
In the text, instead of bonitation score use the visual score.
Answer: Corrected in line with comments.
P6 line 174-175: the data could be presented if even significant differences were not detected.
Answer: For better transparency of work, we prefer to avoid the presentation of additional data
Table 3 Effects of light or salinity?
Answer: Light conditions - Corrected
Page 7 line 201-206: Carefully check the description and the data in Table 5.
Answer: Corrected.
Figure 4. For clarity, the "effects" could be indicated on the picture.
Answer: Done.
Discussion
P10 line 229-236 deals more likely to a review of literature. Please reduce this (or move most in your introduction). The aim of the work should be stated only in the introduction.
Answer: We agree with the reviewer not to repeat the content in the paper, but we wanted to emphasize the originality of the research.
P 11 line 283 specify exactly to which stress.
Answer: The type of stress was determined
Reviewer 2 Report
Dear Authors,
I found that Your research is a bit similar to Rünk work, e.g.
https://dspace.ut.ee/bitstream/handle/10062/1960/rynkkai.pdf?sequence=9&isAllowed=y
Introduction:
50-51 lines: The sentence “However, the effects of light stress on the quality of shade tolerant ornamental plants have not been extensively studied” should be moved to the last paragraph of the Introduction.
MM:
It is not clear how many times the experiment was repeated.
100 line: “plats” to “plants”
106 line: please use only “mmol” or “mM”
125 line: please write “mL” instead of “cm3”
128 line: were three repetitions or three analytical measurements of the same sample?
Results:
137 line: “greenness index” or “chlorophyll index” (SPAD?)
Author Response
Thank you very much for your time spent on a careful and detailed revision of our manuscript. In general, the review let us considerably improve our manuscript. Below you will find our answers to all the remarks. We hope our explanations are comprehensive and will dispel any possible doubts.
Dear Authors,
I found that Your research is a bit similar to Rünk work, e.g.
https://dspace.ut.ee/bitstream/handle/10062/1960/rynkkai.pdf?sequence=9&isAllowed=y
Answer: Thank you for valuable sources of information on the ecology of Dryopteris. we referred to this research at work
Introduction:
50-51 lines: The sentence “However, the effects of light stress on the quality of shade tolerant ornamental plants have not been extensively studied” should be moved to the last paragraph of the Introduction.
Answer: In the last paragraph we emphasize that there is a lack of research on the effect of stress factors on growth and ornamental value of spore plants.
MM:
It is not clear how many times the experiment was repeated.
Answer: The description of the treatments has been improved
100 line: “plats” to “plants”
Answer: This sentence has been deleted
106 line: please use only “mmol” or “mM”
Answer: Only mM was used
125 line: please write “mL” instead of “cm3”
Answer: Done
128 line: were three repetitions or three analytical measurements of the same sample?
Answer: They were three analytical measurements of the same sample.
Results:
137 line: “greenness index” or “chlorophyll index” (SPAD?)
Answer: These terms are synonyms.
Reviewer 3 Report
Overall this is a very sound paper.
Figure 4 would be greatly improved by adding the treatment type above the column of plants, it is difficult to follow with the different treatments only in the figure legend.
Author Response
Thank you very much for your time spent on a careful and detailed revision of our manuscript. In general, the review let us considerably improve our manuscript. Below you will find our answers to all the remarks. We hope our explanations are comprehensive and will dispel any possible doubts.
Overall this is a very sound paper.
Figure 4 would be greatly improved by adding the treatment type above the column of plants, it is difficult to follow with the different treatments only in the figure legend.
Answer: In the pictures of the plants, explanations have been added.